# Numerical Study on the Seismic Response of Fluid-Saturated Porous Media Using the Precise Time Integration Method

**Liang Li [1,*], Shuo Zhou [1], Xiuli Du [1], Jia Song [2] and Chao Gao [1]**

[1] Key Laboratory of Urban Security and Disaster Engineering, Beijing University of Technology, Beijing 100124, China; zhoushuo0726@163.com (S.Z.); duxiuli@bjut.edu.cn (X.D.); gaochaox90@163.com (C.G.)

[2] School of Civil Engineering, North China University of Technology, Beijing 100144, China; sjandrew@163.com

* Correspondence: liliang@bjut.edu.cn; Tel.: +86-10-6739-2430

**Abstract:** The seismic response behavior of fluid-saturated porous media (FSPM) has been a critical subject in the area of soil dynamics and geotechnical earthquake engineering. In this paper, the numerical study of the seismic response of the FSPM is performed based on the *u-p* dynamic formulation. A time-stepping explicit algorithm for the numerical solution to the *u-p* dynamic formulation is developed. The precise time integration method is adopted in the algorithm to improve the computational accuracy. The transmitting artificial boundary is used to describe the energy radiative effect of the wave motion in the FSPM. The numerical results indicate that the time-stepping explicit algorithm developed in the current study is applicable and effective for the numerical solution of the dynamic problems of the FSPM based on the *u-p* dynamic formulation. Furthermore, parametric studies are performed to investigate the effect of the permeability coefficient, elastic modulus of the solid skeleton and porosity on the dynamic response of the FSPM. The analyses show that the permeability coefficient value has a negligible effect on the solid skeleton displacement but has a noticeable impact on the pore fluid pressure. With the decrease of the permeability coefficient value, the peak pore pressure increases remarkably. The elastic modulus of the solid skeleton has an important effect on the solid skeleton displacement and pore fluid pressure. With the decrease of the magnitude of elastic modulus, the solid skeleton displacement and pore fluid pressure increase remarkably. The porosity value has an insignificant effect on the solid skeleton displacement but has a significant impact on the pore fluid pressure. With the increase of the porosity value, the peak pore pressure decreases significantly.

**Keywords:** fluid-saturated porous media; seismic response; *u-p* dynamic formulation; precise time integration method; transmitting artificial boundary

## 1. Introduction

The seismic response behavior of engineering soil field has been the subject of intense research in the area of soil dynamics and geotechnical earthquake engineering for a long time. The numerical results of the seismic response are vital for the seismic safety evaluation of soil fields. On the other hand, the seismic response computation of soils is a necessary step of soil-structure dynamic interaction analysis. Some underground structures are buried in saturated soils, such as immersed tubes in river beds or sea beds and subway tunnels and stations in estuarine deposits or semimetal basins, and it also has practical significance in the computation and analysis of the seismic response of saturated soils.

Saturated soils in nature are fluid-saturated porous media (FSPM). They consist of a solid skeleton and pores. The solid skeleton is composed of soil grains with various shapes. The inter-grain pores are filled with fluid; e.g., water. When the wave motion or seismic response of the FSPM is investigated, the dynamic coupling between the solid skeleton and the pore fluid should be taken into account. The governing equations describing the dynamic behavior of the FSPM were originally proposed by Biot [1] using the solid skeleton displacement $u$ and the pore fluid displacement $U$ as basic variables. Biot [2] also proposed the governing equations for the description of the dynamic behavior of the FSPM using the solid skeleton displacement $u$ and the relative displacement of the pore fluid to the solid skeleton $w$ as basic variables. However, it is more convenient to use the $u$-$p$ dynamic formulation as the governing equations for problems in which high-frequency components are absent, such as those associated with earthquake formulation. The $u$-$p$ dynamic formulation was originally proposed by Zienkiewicz et al. [3] using the solid skeleton displacement $u$ and the pore fluid pressure $p$ as essential variables based on some basic assumptions. As a mixed formulation including the vector and scalar essential variables, the $u$-$p$ dynamic formulation results in a smaller number of nodal unknowns for the finite element formulation in comparison to the $u$-$U$ dynamic formulation including only the vector essential variables. On the other hand, the numerical results of the pore fluid pressure $p$ can be obtained directly from the solution to the $u$-$p$ dynamic formulation, and this is important for some geotechnical earthquake engineering problems, such as soil liquefaction.

The numerical solution procedure for the $u$-$p$ dynamic formulation can be implemented using the time-stepping implicit, semi-explicit or explicit algorithm. For the time-stepping implicit algorithm, the solid skeleton displacement $u$ and pore fluid pressure $p$ are all solved in an implicit mode. This means that a set of coupling linear equations with regard to the solid skeleton displacement $u$ and pore fluid pressure $p$ should be solved in each time step. The time-stepping implicit algorithm is unconditionally stable, but for dynamic problems with a large number of degrees of freedom, the computational effort and computer memory requirement are enormous when it is adopted. Staggered implicit–implicit algorithms for the solution to the $u$-$p$ dynamic formulation have been proposed by Park [4], Zienkiewicz et al. [5], Huang et al. [6,7] and Li et al. [8].

When the semi-explicit algorithm is adopted for the numerical solution to the $u$-$p$ dynamic formulation, one of the two essential variables, such as the solid skeleton displacement, is solved in an implicit mode, while the other variable is solved in an explicit mode. This means that the numerical results of this essential variable can be obtained in an iterative manner, and the computation efficiency is improved to some extent. Zienkiewicz et al. [9] developed two semi-explicit algorithms for the coupled soil skeleton–pore fluid dynamic problem based on the operator splitting before or after the spatial discretization of the dynamic equations. Pastor et al. [10] proposed a time-stepping explicit–implicit algorithm for the numerical solution to the coupled soil skeleton–pore fluid dynamic problem. In this algorithm, the solid skeleton velocity was solved with an explicit computation scheme, and the pore fluid pressure was solved with an implicit computation scheme. Li et al. [11] developed a staggered explicit–implicit algorithm for the numerical solution to the $u$-$p$ dynamic formulation. The solid skeleton displacement was solved in an explicit mode, while the pore fluid pressure was solved with an implicit computation scheme.

In the time-stepping explicit algorithm for the numerical solution to the $u$-$p$ dynamic formulation, both of the two essential variables, solid skeleton displacement $u$ and pore fluid pressure $p$, are solved in an explicit mode. This means that the solid skeleton displacement and pore fluid pressure are all computed in an iterative manner, and the coupled dynamic equations do not need to be solved. The computational effort and computer memory requirement can be reduced significantly by using the explicit algorithm. Song et al. [12] developed a staggered explicit algorithm to calculate the dynamic response of the FSPM based on the $u$-$p$ dynamic formulation.

For the time-domain solution to the dynamic problems of the FSPM, the precise time integration method [13] can be adopted to improve computational accuracy effectively. Duan et al. [14] developed a time-stepping explicit algorithm for the numerical study of the dynamic response of the FSPM.

In this algorithm, the numerical results of the solid skeleton displacement were obtained by the precise time integration method. Du et al. [15] proposed a staggered explicit–implicit algorithm for the numerical solution to the *u-p* dynamic formulation based on the central difference method and precise time integration method. Song et al. [16] developed a staggered explicit–explicit algorithm using the precise time integration method to solve the dynamic problems of the FSPM based on the *u-p* dynamic formulation.

In this paper, the *u-p* dynamic formulation for the description of the dynamic response of the FSPM is introduced, and a time-stepping explicit algorithm based on the precise time integration method is presented. The seismic response of the FSPM is investigated by the proposed algorithm to illustrate the applicability of this algorithm to the numerical solution to the dynamic problems of the FSPM based on the *u-p* dynamic formulation. In the current study, the transmitting artificial boundary [17] is used to describe the energy radiative effect of the wave motion in the FSPM. Parametric studies are also performed to investigate the effect of the permeability coefficient, the elastic modulus of the solid skeleton and porosity on the dynamic response of the FSPM.

## 2. The *u-p* Dynamic Formulation for Wave Motion in FSPM

### 2.1. Basic Assumptions

The following basic assumptions are introduced for the derivation of the *u-p* dynamic formulation for the wave motion in the FSPM:

(1) The solid skeleton is elastic and isotropic;

(2) The deformation of the solid skeleton is small enough that the second term of the derivation of the deformation can be ignored;

(3) The pore fluid is a compressible ideal fluid, and its seepage behavior is governed by Darcy's law;

(4) The volumetric force is ignored;

(5) The pore fluid acceleration relative to the solid skeleton is neglected.

### 2.2. Expressions of u-p Dynamic Formulation

Based on the basic assumptions introduced above, the *u-p* dynamic formulation for the wave motion in the FSPM can be expressed as follows [3]:

$$L^{\mathrm{T}} D L u + L^{\mathrm{T}} m p - \rho \ddot{u} = 0 \tag{1}$$

$$m^{\mathrm{T}} L \dot{u} + \frac{k_f}{\mu_f} \nabla^{\mathrm{T}} \nabla p - \rho_f \frac{k_f}{\mu_f} m^{\mathrm{T}} L \ddot{u} - \frac{n}{E_w} \dot{p} = 0 \tag{2}$$

For the derivation of *u-p* dynamic formulation, the following solid–fluid coupling constitutive relations are adopted from the elastic and isotropic assumption:

$$\sigma' = D e + m p \tag{3}$$

In Equations (1) and (2), the solid skeleton displacement $u$ and pore fluid pressure $p$ are the basic variables which need to be solved. Equation (1) expresses the total momentum balance of the FSPM, and Equation (2) is the flow conservation equation of the pore fluid. $u$, $\dot{u}$ and $\ddot{u}$ are the displacement, velocity and acceleration vectors of the solid skeleton, respectively. For two-dimensional wave motion problems, $u = \{ u_x, \ u_y \}^{\mathrm{T}}$, $\dot{u} = \{ \dot{u}_x, \ \dot{u}_y \}^{\mathrm{T}}$, $\ddot{u} = \{ \ddot{u}_x, \ \ddot{u}_y \}^{\mathrm{T}}$. $\rho$ is the total mass density of the FSPM, $\rho_f$ is the mass density of the pore fluid, $n$ is the porosity, $k_f$ is Darcy's permeability coefficient, $\mu_f$ and $E_w$ are the viscosity coefficient and bulk modulus of the pore fluid, respectively. The gradient operator $\nabla$ and differential operator $L$ are defined as

$$\nabla^{\mathrm{T}} = \left[\begin{array}{cc} \frac{\partial}{\partial x} & \frac{\partial}{\partial y} \end{array}\right], \quad L = \left[\begin{array}{cc} \frac{\partial}{\partial x} & 0 \\ 0 & \frac{\partial}{\partial y} \\ \frac{\partial}{\partial y} & \frac{\partial}{\partial x} \end{array}\right]$$

Vector $m$ can be expressed as

$$m^{\mathrm{T}} = \left[\begin{array}{ccc} 1 & 1 & 0 \end{array}\right]$$

In Equation (3), $\sigma'$ and $e$ are the effective stress and strain vectors of the solid skeleton, respectively. For two-dimensional wave motion problems, $\sigma' = \left\{\begin{array}{ccc} \sigma'_x, & \sigma'_y, \tau'_{xy} \end{array}\right\}^{\mathrm{T}}, e = \left\{\begin{array}{ccc} \varepsilon_x, \varepsilon_y, & \gamma_{xy} \end{array}\right\}^{\mathrm{T}}$. $D$ is the stiffness matrix of the solid skeleton and can be expressed as for the 2D case.

$$D = \left[\begin{array}{ccc} \lambda + 2G & \lambda & 0 \\ \lambda & \lambda + 2G & 0 \\ 0 & 0 & G \end{array}\right]$$

where $\lambda$ and $G$ are the Lame constant and shear modulus of the solid skeleton, respectively. They can be expressed as follows:

$$\lambda = \frac{\nu E_s}{(1+\nu)(1-2\nu)}, \quad G = \frac{E_s}{2(1+\nu)} \tag{4}$$

where $E_s$ and $\nu$ are the elastic modulus and Poisson ratio of the solid skeleton, respectively.

In Equation (2), the term $\rho_f \frac{k_f}{\mu_f} m^{\mathrm{T}} L\ddot{u}$ can be neglected because previous research shows that it has minimum impact on the computation results [18]. By applying the standard Galerkin procedure to Equations (1) and (2), the following finite element discrete expressions can be obtained:

$$M\ddot{u} + Ku + Qp = f_{\mathrm{u}} \tag{5}$$
$$Jp - S\dot{p} + Q^{\mathrm{T}}\dot{u} = f_{\mathrm{p}} \tag{6}$$

where $M$ is the mass matrix, $K$ is the stiffness matrix, $Q$ is the coupling matrix between the solid skeleton and the pore fluid, $J$ is the seepage matrix of the pore fluid, $S$ is the compression matrix of the pore fluid, and $f_u$ and $f_p$ are the boundary force vectors acting on the solid skeleton and pore fluid, respectively. The matrices in Equations (5) and (6) can be expressed as

$$M = \rho \int_\Omega N_u{}^{\mathrm{T}} N_u \mathrm{d}\Omega,$$
$$K = \int_\Omega (LN_u)^{\mathrm{T}} D(LN_u) \mathrm{d}\Omega,$$
$$Q = \int_\Omega (LN_u)^{\mathrm{T}} m(LN_u) \mathrm{d}\Omega,$$
$$S = \frac{E_w}{n} \int_\Omega N_p{}^{\mathrm{T}} N_p \mathrm{d}\Omega,$$
$$J = \frac{k_f}{\mu_f} \int_\Omega (\nabla N_p)^{\mathrm{T}} (\nabla N_p) \mathrm{d}\Omega,$$

where $N_u$ and $N_p$ are the shape functions for the solid skeleton displacement $u$ and pore fluid pressure $p$, respectively.

The stepwise governing equations at time step $k$ can be written as follows

$$M\ddot{u}_k + Ku_k + Qp_k = f_{\mathrm{u}k} \tag{7}$$

$$Jp_k - S\dot{p}_k + Q^{\mathrm{T}}\dot{u}_k = f_{\mathrm{p}k} \tag{8}$$

## 3. Time-Stepping Explicit Algorithm for Numerical Solution to *u-p* Dynamic Formulation

*3.1. Algorithm Implementation*

Equation (5) can be rewritten as follows:

$$M\ddot{u} + C\dot{u} + Ku = F \tag{9}$$

where $C$ is the zero matrix, and $F = f_u - Qp$.

Let

$$y = M\dot{u} + C u/2 \tag{10}$$

From Equation (10), we can obtain

$$\dot{u} = M^{-1}y - M^{-1}Cu/2 \tag{11}$$

Substituting Equation (11) into Equation (9) results in

$$\dot{y} = F(t) - (K - CM^{-1}C/4)u - CM^{-1}y/2 \tag{12}$$

Combing Equations (11) and (12) results in the following equations:

$$\dot{v} = Hv + r \tag{13}$$

where

$$v = \begin{bmatrix} u \\ y \end{bmatrix}, \; H = \begin{bmatrix} A & D \\ B & G \end{bmatrix}, \; r = \begin{bmatrix} 0 \\ F(t) \end{bmatrix}$$
$$A = M^{-1}C/2, \; G = -CM^{-1}/2, \; B = K - CM^{-1}C/4, \; D = M^{-1}$$

Equation (13) is the equivalent form of the *u-p* dynamic formulation for the wave motion in the FSPM. The solution of Equation (13) consists of the general solution of the homogeneous equation and the particular solution of the nonhomogeneous equation. The general solution can be expressed as

$$v(t) = e^{Ht}v_0 + \int_0^t e^{H(t-\tau)}r(\tau)\mathrm{d}\tau \tag{14}$$

From Equation (14), we can obtain the following expression of the iterative relation for the dynamic response at the moment $t_{k+1}$ and $t_k$.

$$v(t_{k+1}) = e^{H\Delta t}v(t_k) + \int_{t_k}^{t_{k+1}} e^{H(t_{k+1}-\tau)}r(\tau)\mathrm{d}\tau \tag{15}$$

The integral term in Equation (15) can be computed with the Gaussian integral method as follows:

$$\int_{t_k}^{t_{k+1}} e^{H(t_{k+1}-\tau)}r(\tau)\mathrm{d}\tau = \frac{\Delta t}{2}\sum_{i=1}^{n}\omega_i T(\frac{\Delta t}{2}(1-\xi_i))r(t_k + \frac{\Delta t}{2}(1+\xi_i)) + o(\Delta t^{2n+1}) \tag{16}$$

where $n$ is the number of integral points, $\xi_i$ is the coordinates of integral points, and $\omega_i$ is the weighted coefficient.

The exponential matrix on the right of Equation (15) can be computed by the precise time integration method as follows:

$$T(\Delta t) = e^{H\Delta t} = (e^{H\Delta t/2^N})^{2^N} = (e^{H\tau})^{2^N} \tag{17}$$

Substituting Equations (16) and (17) into Equation (15) results in the expression of the dynamic response at the moment $t_{k+1}$:

$$v(t_{k+1}) = T(\Delta t)v(t_k) + \frac{\Delta t}{2}\sum_{i=1}^{n}\omega_i T(\frac{\Delta t}{2}(1-\xi_i))r(t_k + \frac{\Delta t}{2}(1+\xi_i)) + o(\Delta t^{2n+1}) \tag{18}$$

where $r(t_k + \frac{\Delta t}{2}(1 + \xi_i))$ is related to the pore fluid pressure at the moment $t_k$ and its derivative to time $p_k$ and $\dot{p}_k$.

It is assumed that the pore fluid pressure varies linearly in the time interval $\Delta t$; then, its derivative to time can be computed by a difference method. When the backward difference method is applied, we can obtain

$$\dot{p}_k = (p_k - p_k)/\Delta t \tag{19}$$

Substituting Equation (19) into Equation (8) results in the iterative expression of the pore fluid pressure at the moment $t_{k+1}$ as follows:

$$p_{k+1} = p_k + \Delta t S^{-1}(f_p - J p_k - Q^{\mathrm{T}} \dot{u}_k) \tag{20}$$

Equations (18) and (20) consist of a time-stepping explicit algorithm for the numerical solution of the *u-p* dynamic formulation. Using these equations, the dynamic response of the FSPM, including the solid skeleton displacement and pore fluid pressure, can be calculated in an iterative way. This algorithm does not need to solve coupled dynamic equations at each time step, so the computational efficiency can be improved remarkably.

### 3.2. Algorithm Validation

To validate the algorithm, the numerical results from the algorithm are compared with the analytical solution from Simon et al. [19]. The calculation model simulates a saturated soil deposit subject to a sinusoidal surface load. Figure 1 shows the calculation model including the geometries and boundary conditions. The dimension of the calculation model is 3 m by 30 m. The top surface of the model is a free surface, the bottom surface is a fixed in both the horizontal and vertical direction, and the left and right sides surfaces are fixed in the horizontal direction. All the boundaries are impermeable, except the top surface. The time history of the applied load is shown in Figure 2. The comparison of the numerical results with the analytical solution is shown in Figure 3. The non-dimensional time $\tau$ and displacement $\hat{u}$ are used for the comparison. The good agreement between the numerical results and the analytical solution can be illustrated by the comparison presented in Figure 3.

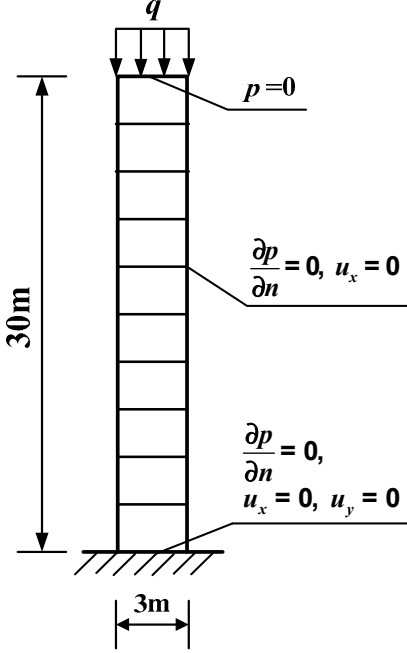

**Figure 1.** Calculation model for the algorithm validation.

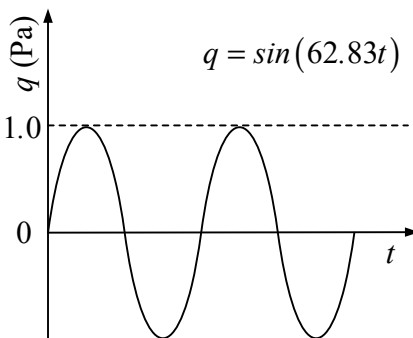

**Figure 2.** Time history of the applied load.

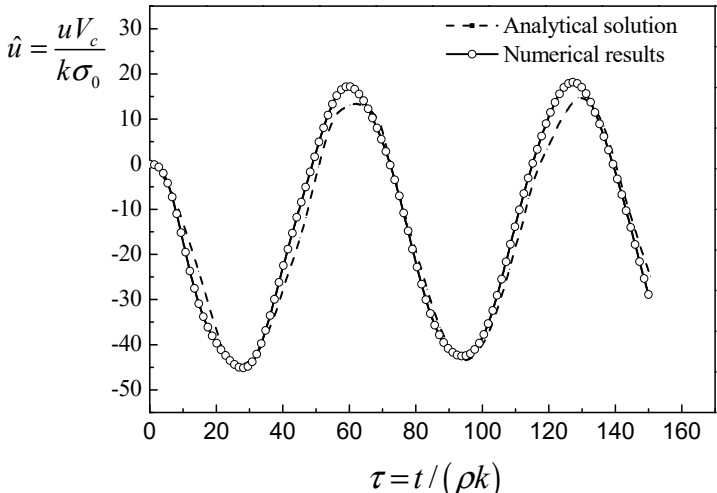

**Figure 3.** Comparison of the numerical results with the analytical solution.

## 4. Numerical Study on Seismic Response of FSPM

In this section, the time-stepping explicit algorithm derived above is applied to study the dynamic response of the FSPM in a rectangular domain under seismic loadings. The transmitting artificial boundary [17] is applied to simulate the wave propagation toward the far field infinite domain. The calculation model is shown in Figure 4. The dimension of the calculating domain is 80 m by 80 m. The top surface of the calculation model is set as the free and drainage boundary, and the left, right and bottom boundaries of the calculation model are all set as the transmitting artificial boundary. The four-node rectangular element is used for the finite element discretization of the calculation domain. Although high computational accuracy can be gained using a relatively small mesh size, this will lead to an increase of the number of finite element meshes. The computational effort and computer memory required will be enormous. From the associated consideration for the computational accuracy and effort, a finite element mesh size of 8 m by 8 m is adopted for the numerical study in this section. The strong motion record from the Loma Prieta earthquake is used as seismic excitation, input perpendicularly from the bottom boundary of the finite element model as a compressive wave and shear wave, respectively. The duration of the Loma Prieta earthquake record is 48.7 s; the displacement and velocity time history of the strong motion record are shown in Figures 5 and 6. The material properties of the FSPM are shown in Table 1. The time step $\Delta t$ used for the calculation is 0.0001 s.

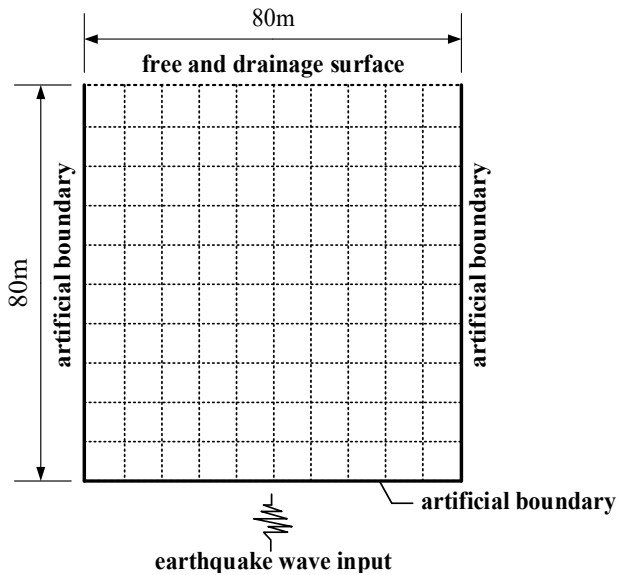

**Figure 4.** Calculation model for the seismic response of the FSPM.

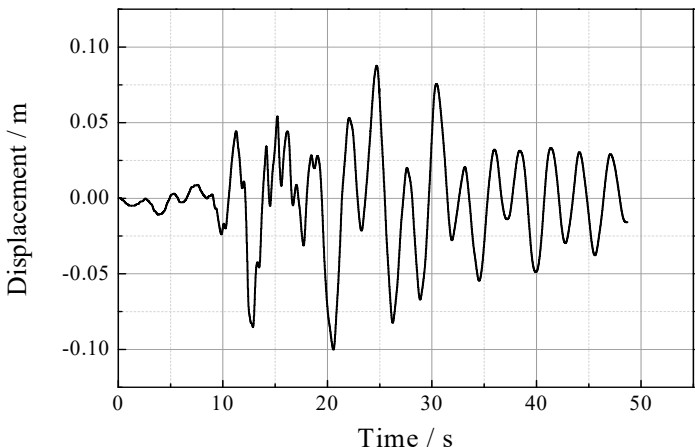

**Figure 5.** Displacement time history of the Loma Prieta earthquake record.

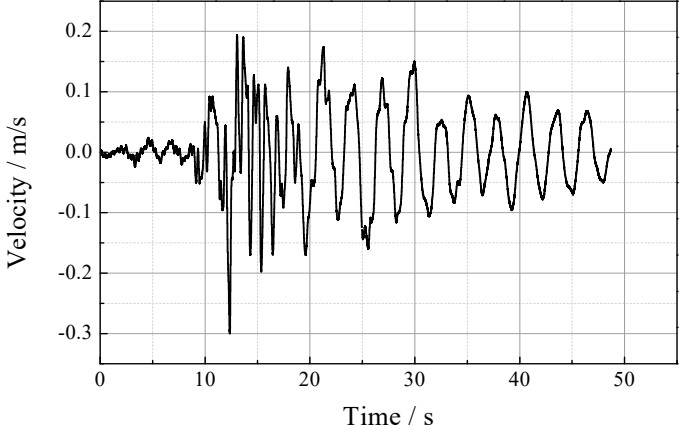

**Figure 6.** Velocity time history of the Loma Prieta earthquake record.

**Table 1.** Material properties of the fluid-saturated porous media (FSPM).

| $\lambda$ | $G$ | $E_w$ | $\rho_f$ | $\rho$ | $k_f$ | $n$ |
|:---:|:---:|:---:|:---:|:---:|:---:|:---:|
| (Pa) | (Pa) | (Pa) | (kg/m$^3$) | (kg/m$^3$) | (m/s) | |
| $8.33 \times 10^6$ | $1.25 \times 10^7$ | $1.0 \times 10^5$ | 1000 | 1700 | $1.0 \times 10^{-2}$ | 0.3 |

The dynamic response time histories of the FSPM calculating domain when seismic excitation is input as a compressive wave are illustrated in Figure 7 to Figure 8, and the dynamic response time histories when seismic excitation is input as a shear wave are illustrated in Figure 9 to Figure 10. For the compressive wave, the direction for particle vibration is in accordance with the direction of the wave spread, and so the displacement and velocity time history of the solid skeleton of the free surface in the vertical direction are presented in Figures 7 and 11, respectively. For the shear wave, the direction for particle vibration is perpendicular to the direction of the wave spread, so the displacement and velocity time history of the solid skeleton of the free surface in the horizontal direction are presented in Figures 9 and 12, respectively. The pore fluid pressure time history of the central depth of the calculating domain for the compressive wave and shear wave input are presented in Figures 8 and 10, respectively, since the drainage condition has been assumed for the top surface of the calculating domain.

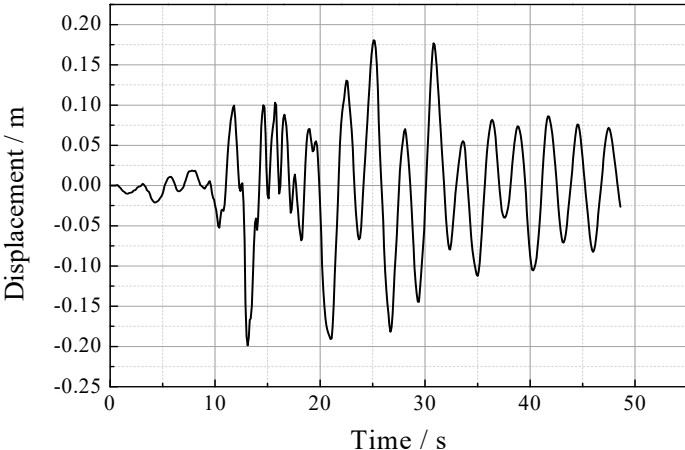

**Figure 7.** Solid skeleton displacement time history of the free surface in the vertical direction (compressive wave input).

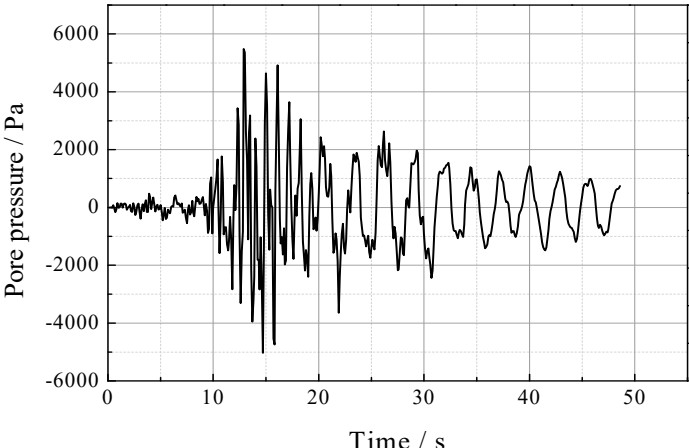

**Figure 8.** Pore fluid pressure time history of the central depth of the calculating domain (compressive wave input).

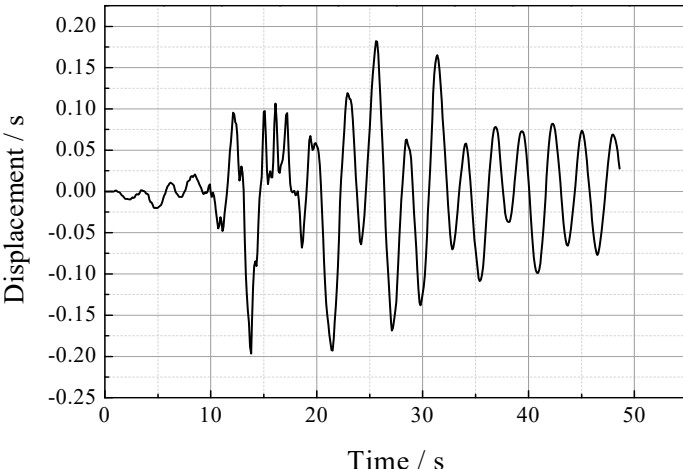

**Figure 9.** Solid skeleton displacement time history of the free surface in the horizontal direction (shear wave input).

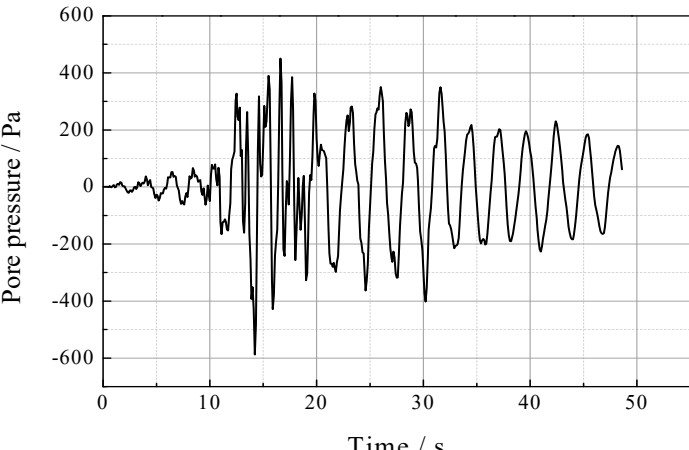

**Figure 10.** Pore fluid pressure time history of the central depth of the calculating domain (shear wave input).

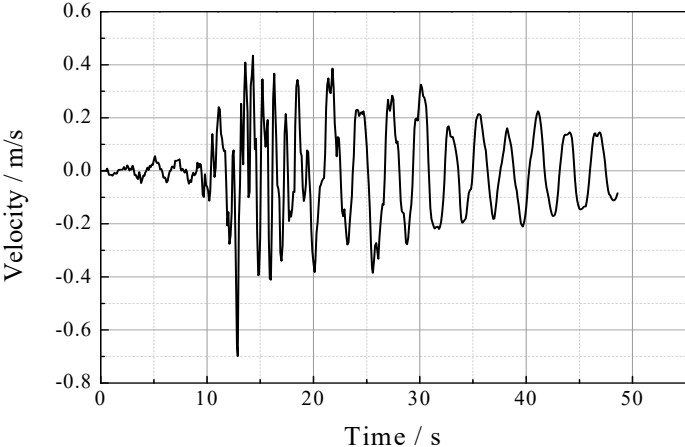

**Figure 11.** Solid skeleton velocity time history of the free surface in the vertical direction (compressive wave input).

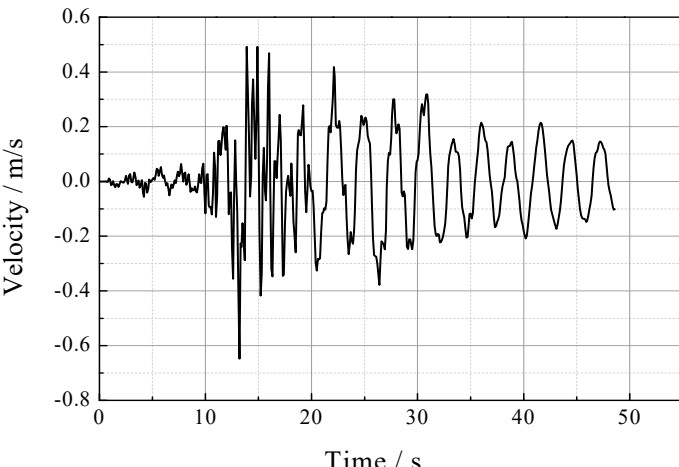

**Figure 12.** Solid skeleton velocity time history of the free surface in the horizontal direction (shear wave input).

It is shown in Figures 7 and 9 that, when the seismic excitation is input perpendicularly from the bottom boundary of calculating domain, the profiles of the solid skeleton displacement time history of the free surface of the calculating domain are in accordance with that of the displacement time history of the incident seismic motion, and the peak displacement of the free surface has a roughly two-fold increase relative to the peak displacement of the incident seismic record. A similar trend can be observed for the solid skeleton velocity time history of the free surface of the calculating domain from Figures 11 and 12. The numerical results presented above are in accordance with the elastic wave theories [20]. This indicates that the time-stepping explicit algorithm proposed in the current study is applicable and effective for the numerical solution of the dynamic problems of the FSPM based on the *u-p* dynamic formulation.

## 5. Sensitivity of the Material Properties of the FSPM

In this section, parametric studies are performed to study the sensitivity of the dynamic response of the FSPM to the material properties, including the permeability coefficient, elastic modulus of the solid skeleton and porosity. A new calculation model, shown in Figure 13, is used for the study. The width and height of the calculating domain are 100 m and 50 m, respectively. The top surface of the calculation model is set as the free and drainage boundary, and the left, right and bottom boundaries of the calculation model are all set as the transmitting artificial boundary. The four-node rectangular element is used for the finite element discretization of the calculation domain, From the associated consideration of the computational accuracy and effort, a finite element mesh size of 5 m by 5 m is adopted for the numerical study in this section. The dynamic response of node A (50, 45) and B (50, 20) in the finite element model are investigated.

The strong motion record from the Ninghe earthquake is used as seismic excitation, input perpendicularly from the bottom boundary of the finite element model as the compressive wave. The duration of the Ninghe earthquake record is 19.2 s; the displacement time history of the strong motion record is shown in Figure 14. The time step $\Delta t$ used for the calculation is 0.0001 s.

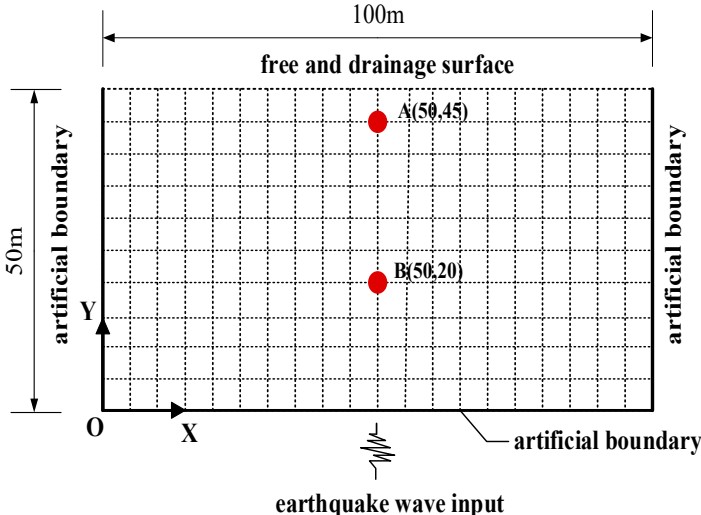

**Figure 13.** Calculation model for the parametric study of the seismic response of the FSPM.

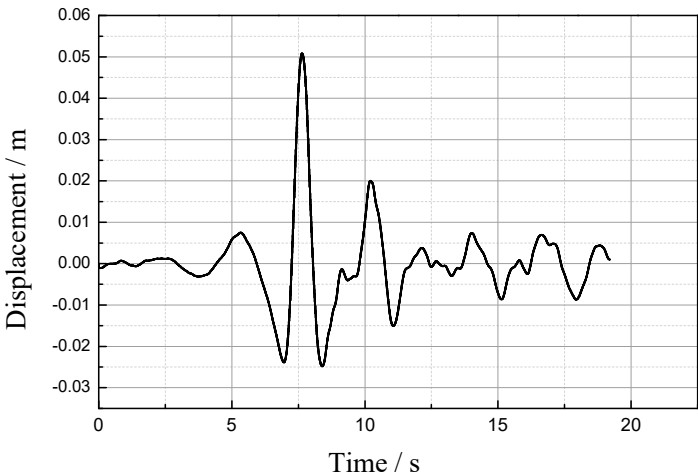

**Figure 14.** Displacement time history of the Ninghe earthquake record.

### 5.1. Sensitivity of the Permeability Coefficient

The material parameters of the FSPM for the sensitivity analysis of the permeability coefficient are listed in Table 2. The solid skeleton displacement time histories of nodes A and B with different permeability coefficient values are presented in Figures 15 and 16, respectively. These figures show that the peak displacement of nodes A and B reveal negligible variety when the permeability coefficient varies from $10^{-3}$ m/s to $10^{-5}$ m/s. This indicates that the permeability coefficient value has an insignificant effect on the solid skeleton displacement. The horizontal distribution of the solid skeleton displacement in the depth of nodes A and B at a certain moment (t = 10 s) for different permeability coefficients are presented in Figures 17 and 18, respectively. The vertical distribution of the solid skeleton displacement at a certain moment for different permeability coefficients is shown in Figure 19. An increase of the magnitude of solid skeleton displacement in the central section of the horizontal position can be observed with the decrease of the permeability coefficient value, but negligible variety is revealed for the vertical distribution of solid skeleton displacement. The pore fluid pressure time histories of nodes A and B with different permeability coefficient values are presented in Figures 20 and 21, respectively. It is shown that with the decrease of the magnitude of permeability coefficient, the peak pore pressure increases remarkably. When the permeability coefficient varies from $10^{-3}$ m/s to $10^{-5}$ m/s, the peak pore pressure of node A increases by about 35%. This indicates that the permeability coefficient value has a noticeable effect on the pore fluid pressure. The permeability coefficient is a

parameter used to characterize the permeability and drainage speed of the FSPM. A small permeability coefficient value will lead to a slow drainage speed and produce an accumulation of pore fluid pressure in the FSPM more easily.

**Table 2.** Material parameters of the FSPM for the sensitivity analysis of the permeability coefficient.

| $\lambda$ | $G$ | $E_w$ | $\rho_f$ | $\rho$ | $k_f$ | | $n$ |
|-----------|------|-------|----------|--------|-------|---|-----|
| (Pa) | (Pa) | (Pa) | (kg/m³) | (kg/m³) | (m/s) | | |
| $8.33 \times 10^6$ | $1.25 \times 10^7$ | $1.0 \times 10^5$ | 1000 | 1700 | $1.0 \times 10^{-3}, 1.0 \times 10^{-4}, 1.0 \times 10^{-5}$ | | 0.3 |

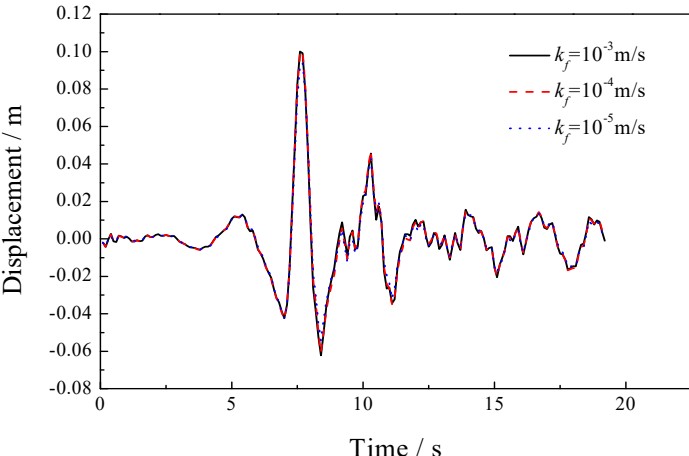

**Figure 15.** Solid skeleton displacement time history of node A for different permeability coefficients.

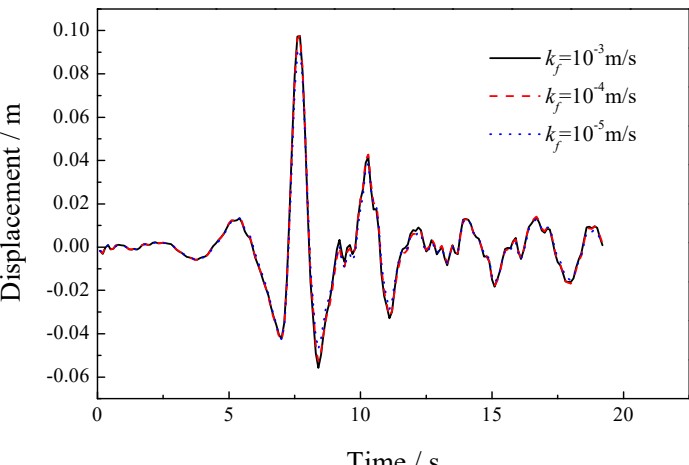

**Figure 16.** Solid skeleton displacement time history of node B for different permeability coefficients.

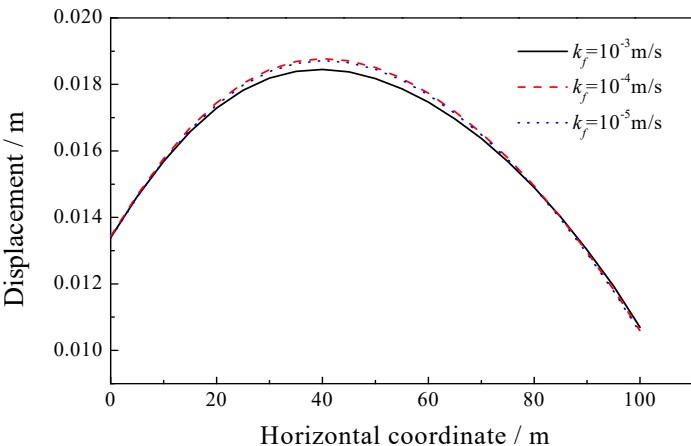

**Figure 17.** Horizontal distribution of solid skeleton displacement for different permeability coefficients (y = 45 m, t = 10 s).

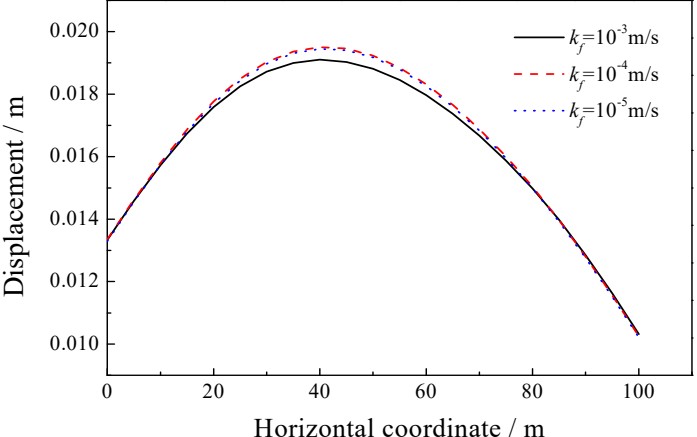

**Figure 18.** Horizontal distribution of solid skeleton displacement for different permeability coefficients (y = 20 m, t = 10 s).

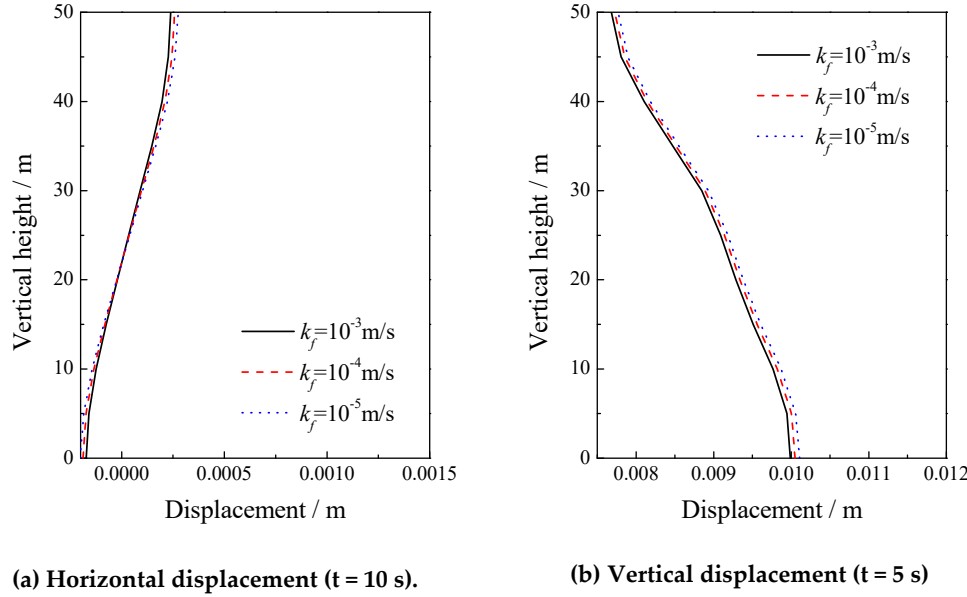

**(a) Horizontal displacement (t = 10 s).**　　　　**(b) Vertical displacement (t = 5 s)**

**Figure 19.** Vertical distribution of solid skeleton displacement for different permeability coefficients.

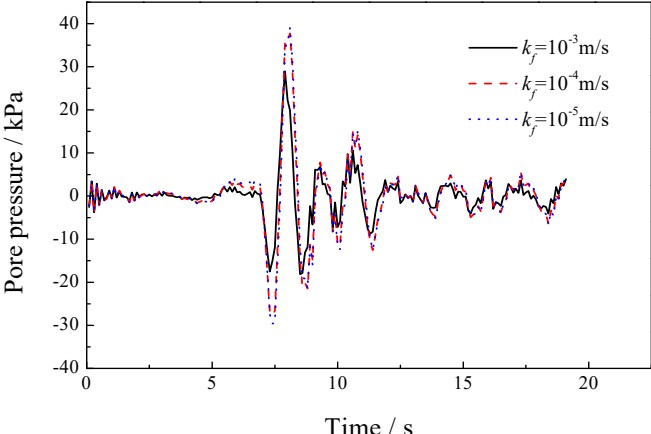

**Figure 20.** Pore fluid pressure time history of node A for different permeability coefficients.

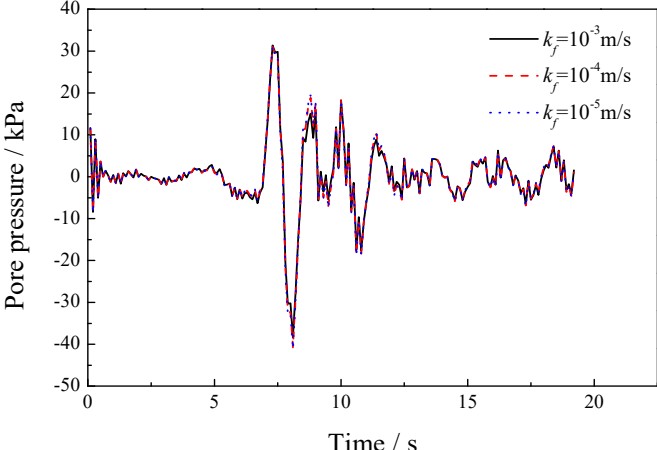

**Figure 21.** Pore fluid pressure time history of node B for different permeability coefficients.

## 5.2. Sensitivity of the Elastic Modulus of the Solid Skeleton

The material parameters of the FSPM for the sensitivity analysis of the elastic modulus of the solid skeleton are shown in Table 3. The solid skeleton displacement time histories of nodes A and B with different elastic modulus values are presented in Figures 22 and 23, respectively. It is observed that with the increase of the magnitude of the elastic modulus, the peak value of solid skeleton displacement decreases noticeably. When the elastic modulus varies from $2.4 \times 10^7$ Pa to $3.6 \times 10^7$ Pa, the peak value of the solid skeleton displacement of node A decreases by about 20%. This shows that the elastic modulus of the solid skeleton has a noticeable effect on the solid skeleton displacement. The horizontal distribution of the solid skeleton displacement in the depth of nodes A and B at a certain moment (t = 10 s) for different elastic modulus are presented in Figures 24 and 25, respectively. The vertical distribution of the solid skeleton displacement at a certain moment for different elastic modulus is shown in Figure 26. An increase of the magnitude of solid skeleton displacement along the horizontal position and vertical height can be observed with the decrease of the elastic modulus value. The pore fluid pressure time histories of nodes A and B with different elastic modulus values are presented in Figures 27 and 28, respectively. It is shown that with the decrease of the magnitude of elastic modulus, the peak pore pressure increases remarkably. When the elastic modulus varies from $3.6 \times 10^7$ Pa to $2.4 \times 10^7$ Pa, the peak pore pressure of node B increases by about 42%. This indicates that the elastic modulus of the solid skeleton has a remarkable effect on the pore fluid pressure. A small elastic modulus value will lead to a larger volume deformation of solid skeleton; this will produce a higher accumulation of pore fluid pressure due to the dynamic coupling between the solid skeleton and pore fluid.

**Table 3.** Material parameters of the FSPM for the sensitivity analysis of the elastic modulus.

| $E_s$ | $\nu$ | $E_w$ | $\rho_f$ | $\rho$ | $k_f$ | $n$ |
|---|---|---|---|---|---|---|
| (Pa) | | (Pa) | (kg/m$^3$) | (kg/m$^3$) | (m/s) | |
| $2.4 \times 10^7, 3.0 \times 10^7, 3.6 \times 10^7$ | 0.2 | $1.0 \times 10^5$ | 1000 | 1700 | $1.0 \times 10^{-2}$ | 0.3 |

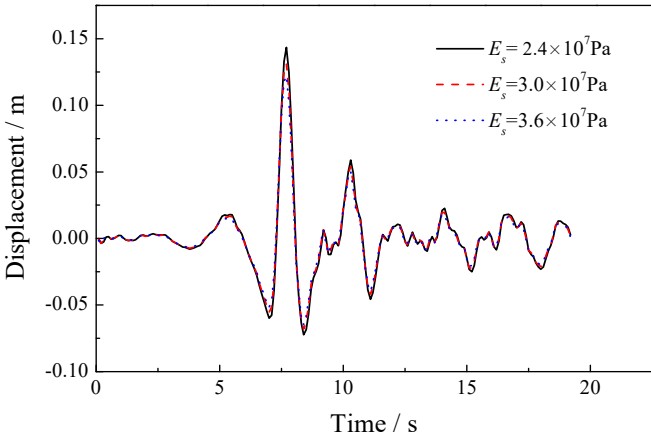

**Figure 22.** Solid skeleton displacement time history of node A for different elastic modulus.

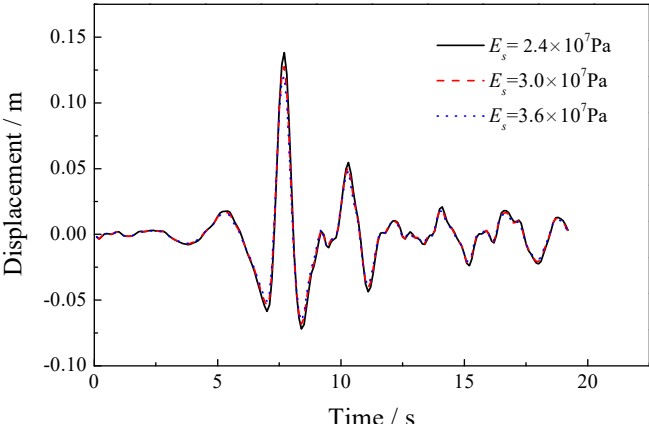

**Figure 23.** Solid skeleton displacement time history of node B for different elastic modulus.

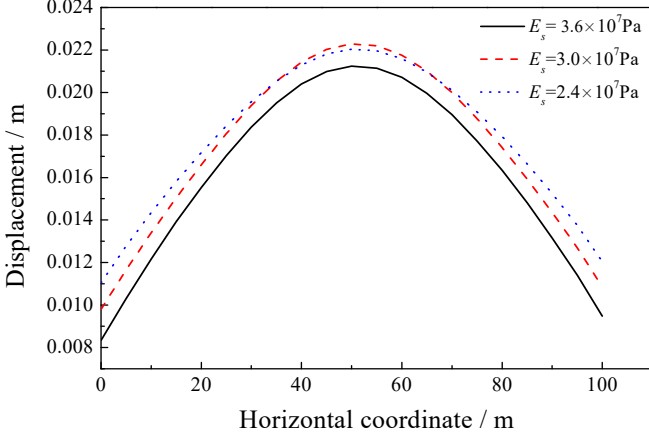

**Figure 24.** Horizontal distribution of solid skeleton displacement for different elastic modulus (y = 45 m, t = 10 s).

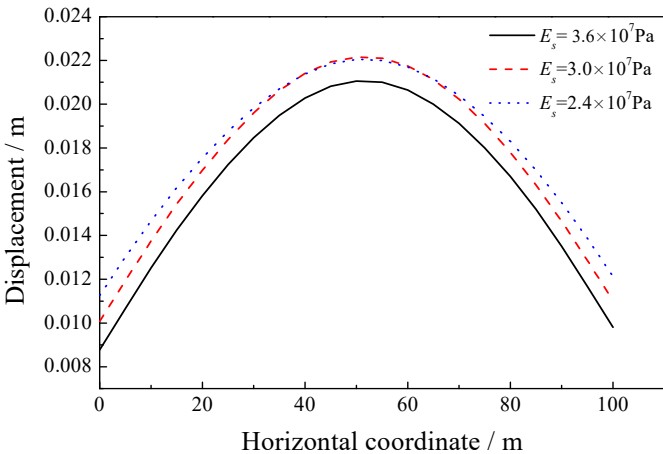

**Figure 25.** Horizontal distribution of solid skeleton displacement for different elastic modulus (y = 20 m, t = 10 s).

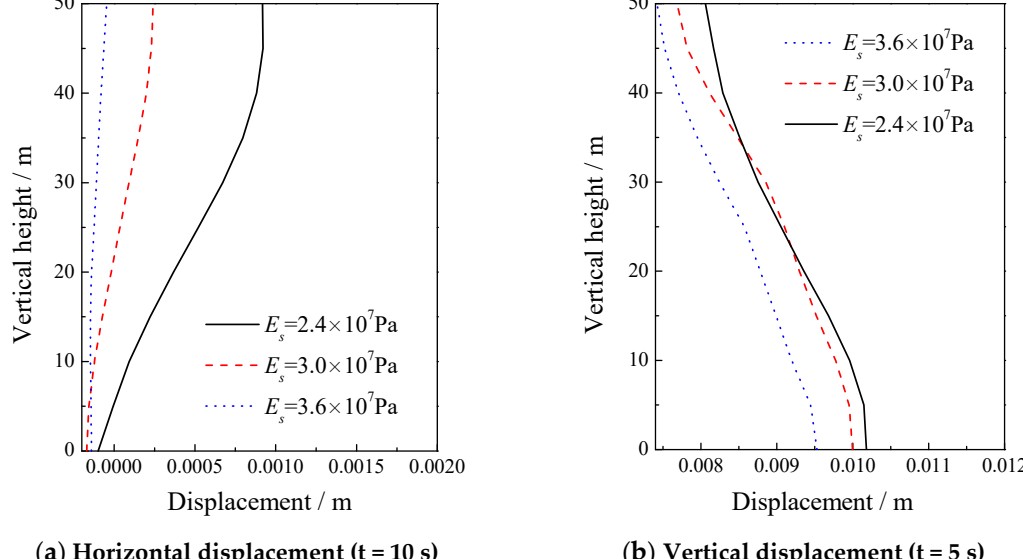

(**a**) **Horizontal displacement (t = 10 s)**          (**b**) **Vertical displacement (t = 5 s)**

**Figure 26.** Vertical distribution of solid skeleton displacement for different elastic modulus.

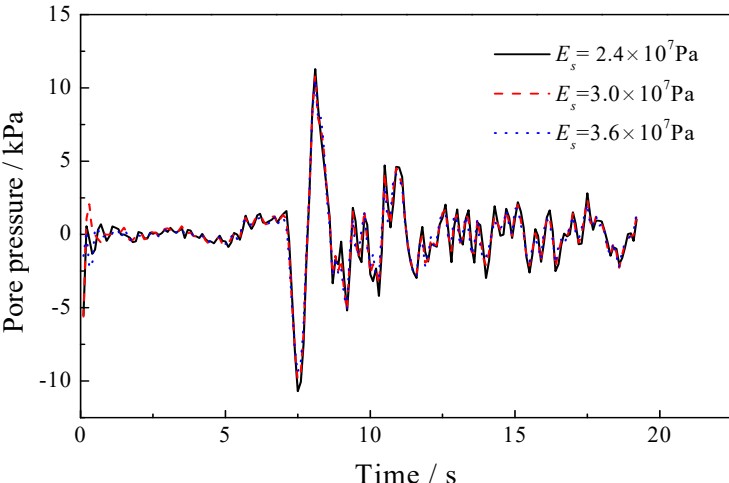

**Figure 27.** Pore fluid pressure time history of node A for different elastic modulus.

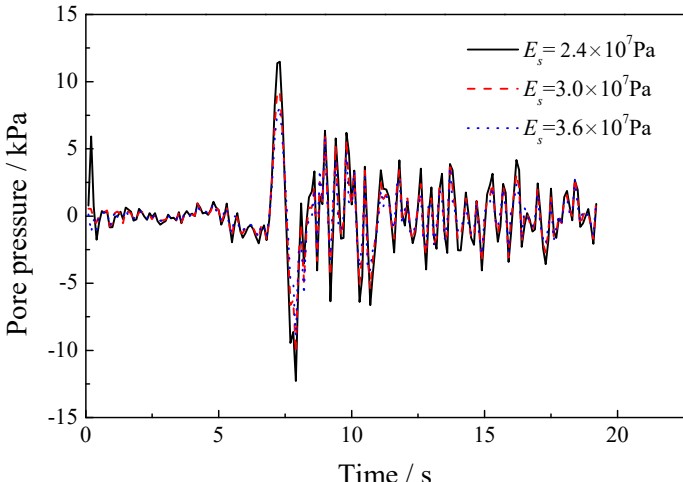

**Figure 28.** Pore fluid pressure time history of node B for different elastic modulus.

## 5.3. Sensitivity of the Porosity

The material parameters of the FSPM for the sensitivity analysis of the porosity are shown in Table 4. The solid skeleton displacement time histories of nodes A and B with different porosity values are presented in Figures 29 and 30, respectively. These figures show that the peak displacements of nodes A and B reveal negligible variety when the porosity varies from 0.2 to 0.4. This indicates that the porosity value has an insignificant effect on the solid skeleton displacement. The horizontal distribution of the solid skeleton displacement in the depth of nodes A and B at a certain moment (t = 10 s) for different porosities are presented in Figures 31 and 32, respectively. The vertical distribution of the solid skeleton displacement at a certain moment for different porosities is shown in Figure 33. An increase of the magnitude of solid skeleton displacement along the horizontal position and vertical height can be found with the increase of porosity value. The pore fluid pressure time histories of nodes A and B with different porosity values are presented in Figures 34 and 35, respectively. It is shown that with the increase of the porosity value, the peak pore pressure decreases remarkably. When the porosity varies from 0.2 to 0.4, the peak pore pressure of node A decreases by about 40%, and the peak pore pressure of node B decreases by about 51%. This indicates that the porosity value has a significant effect on the pore fluid pressure.

**Table 4.** Material parameters of the FSPM for the sensitivity analysis of the porosity.

| $\lambda$ | $G$ | $E_w$ | $\rho_f$ | $\rho$ | $k_f$ | $n$ |
|---|---|---|---|---|---|---|
| (Pa) | (Pa) | (Pa) | (kg/m$^3$) | (kg/m$^3$) | (m/s) | |
| $8.33 \times 10^6$ | $1.25 \times 10^7$ | $1.0 \times 10^5$ | 1000 | 1700 | $1.0 \times 10^{-2}$ | 0.2, 0.3, 0.4 |

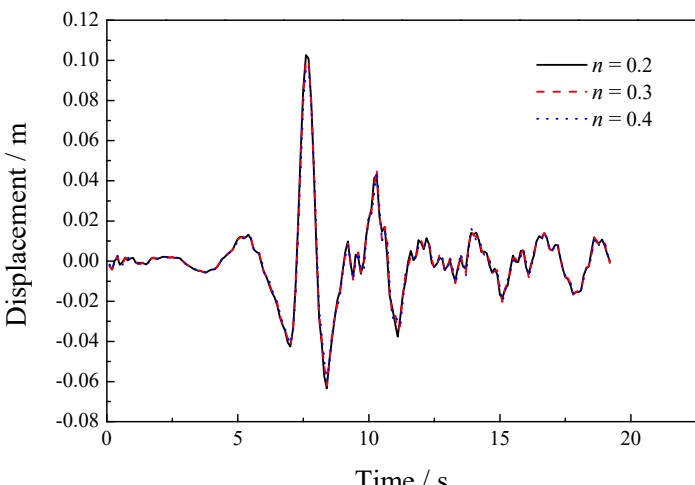

**Figure 29.** Solid skeleton displacement time history of node A for different porosities.

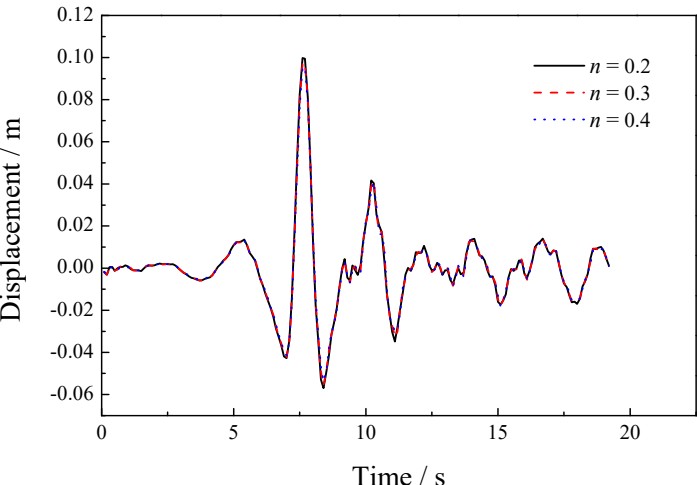

**Figure 30.** Solid skeleton displacement time history of node B for different porosities.

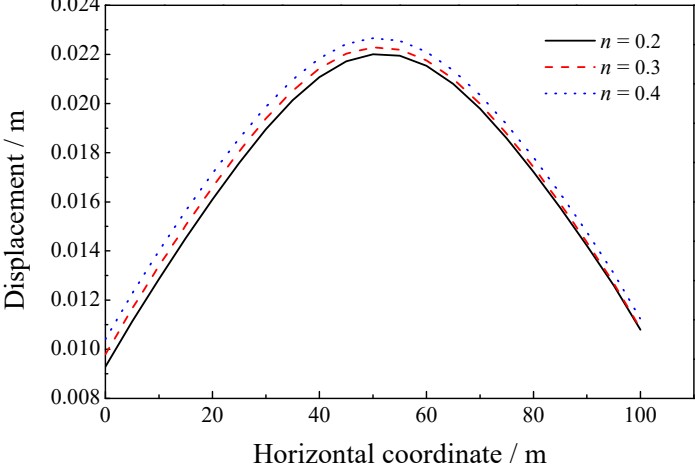

**Figure 31.** Horizontal distribution of solid skeleton displacement for different porosities (y = 45 m, t = 10 s).

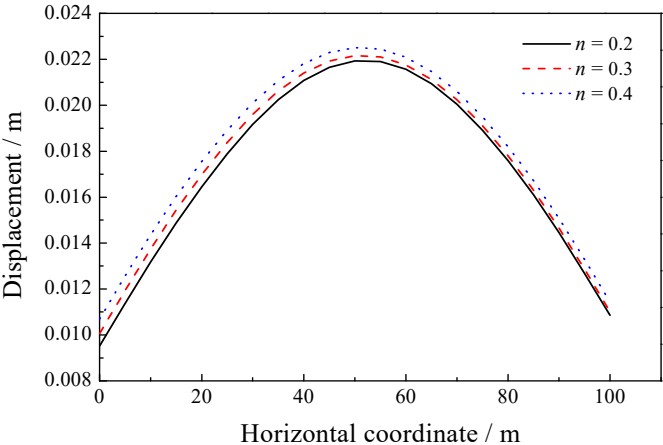

**Figure 32.** Horizontal distribution of solid skeleton displacement for different porosities (y = 20 m, t = 10 s).

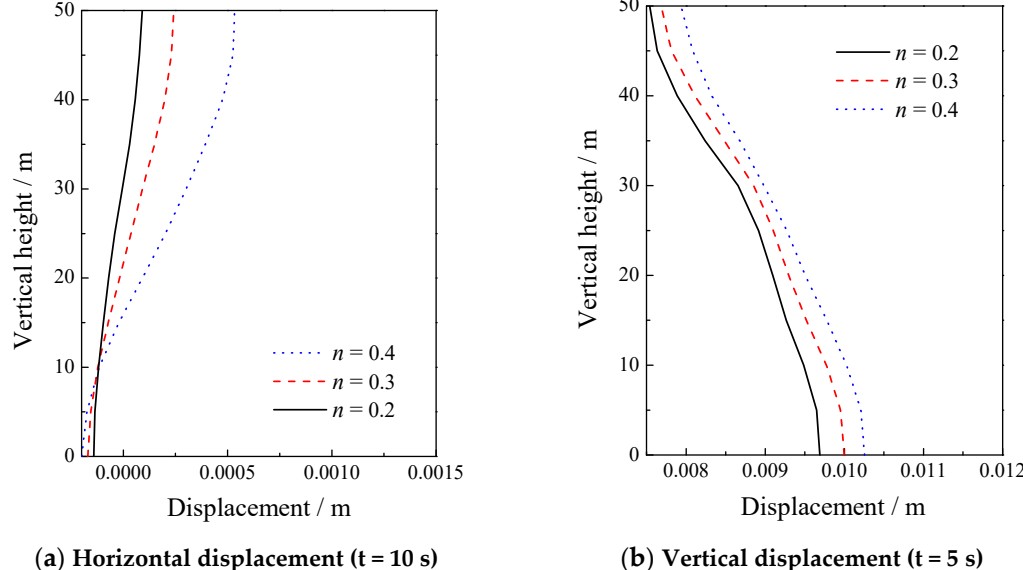

(**a**) **Horizontal displacement (t = 10 s)**          (**b**) **Vertical displacement (t = 5 s)**

**Figure 33.** Vertical distribution of solid skeleton displacement for different porosities.

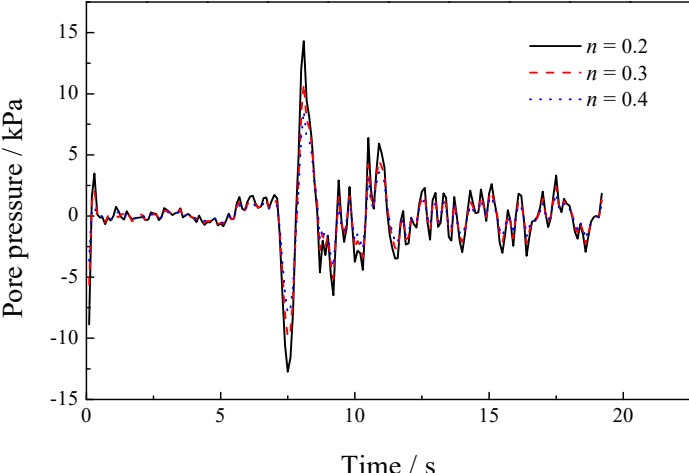

**Figure 34.** Pore fluid pressure time history of node A for different porosities.

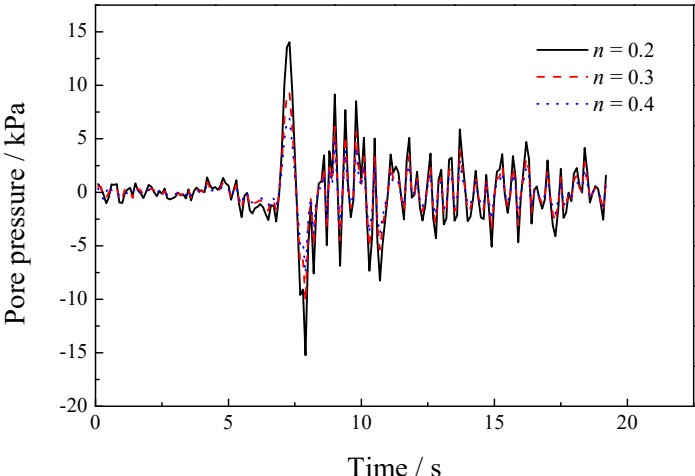

**Figure 35.** Pore fluid pressure time history of node B for different porosities.

## 6. Conclusions

In this paper, a numerical study of the seismic response of the FSPM is performed. The *u-p* dynamic formulation for the description of the dynamic response of the FSPM is introduced. A time-stepping explicit algorithm for the numerical solution of the *u-p* dynamic formulation is developed. The precise time integration method is adopted in the algorithm to improve the computational accuracy. The proposed algorithm is then applied to investigate the seismic response of the FSPM. The transmitting artificial boundary is used to describe the energy radiative effect of the wave motion in the FSPM. The corresponding numerical results obtained by the algorithm are in accordance with the elastic wave theories. This indicates that the time-stepping explicit algorithm developed in the current study is applicable and effective for the numerical solution of the dynamic problems of the FSPM based on the *u-p* dynamic formulation. Furthermore, parametric studies are performed to investigate the effect of the permeability coefficient, elastic modulus of the solid skeleton and porosity on the dynamic response of the FSPM. The analyses show that the permeability coefficient value has a negligible effect on the solid skeleton displacement but has a noticeable impact on the pore fluid pressure. With the decrease of the magnitude of the permeability coefficient, the peak pore pressure increases remarkably. The elastic modulus of the solid skeleton has an important effect on the solid skeleton displacement and pore fluid pressure. With the decrease of the magnitude of the elastic modulus, the solid skeleton displacement and pore fluid pressure increase remarkably. The porosity value has an insignificant effect on the solid skeleton displacement but has a significant impact on the pore fluid pressure. With the increase of the porosity value, the peak pore pressure decreases significantly.

**Author Contributions:** L.L. developed the time-stepping explicit numerical algorithm and performed the formula derivation for the algorithm. L.L. prepared the paper. S.Z. performed the program composition for the algorithm and the numerical computation of the seismic response of fluid-saturated porous media. X.D. discussed the numerical results and provided suggestions on the overall organization of the paper. J.S. performed the parametric studies. C.G. revised the manuscript.

**Funding:** This research was funded by the National Natural Science Foundation of People's Republic of China (No. 51421005 and 51808006). The financial support from both foundations is gracefully acknowledged.

**Conflicts of Interest:** The authors declare no conflict of interest.

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
