# Peer review of "Numerical Study on the Seismic Response of Fluid-Saturated Porous Media Using the Precise Time Integration Method"

_applsci, doi:10.3390/app9102037_

Round 1
Reviewer 1 Report
The paper presents a numerical study on the seismic response of fluid saturated porous media using the precise time integration method through a sensitivity analysis.
In the sections 1, 2 and 3, the presentation of background and the dynamic formulation is concise and clear.
Not so clear and effective is the second part of the paper concerning the numerical study on seismic response of FSPM (cection 4) using Loma Pietra earthquake record and sensitivity analysis (section 5) using Ninghe earthquake record.
a) In section 4, it is used a square domain (80m, 80m) for a discretization considering a size of the finite element meshes of 8m by 8m (Loma Pietra earthquake record).
In section 5, it is used a rectangular domain (100m, 50m) for a discretization considering a size of the finite element meshes of 5m by 5m (Ninghe earthquake record).
If these choices had been made as a consequence of an optimization process, this process should be presented in the paper.
b) In the section 4 (table 1) the material properties used in the analyses are presented. In the section 5, the material properties used in the analyses are not exhaustively presented. They should be fully presented.
The sensitivity analysis had considered only the variability of permeability coefficient and of porosity assuming constant values of the Lame constant and of shear modulus on the basic assumptions of section 2.1 (compressibility of pore fluid, etc…). The pore fluid pressure time history can be strongly influenced by variability of other parameters. Even these choices should be presented.
c) The analyses results are not presented exhaustively, only time history of two points of domain are selected. You could use a more effective representation of the results (for example: vertical and horizontal profiles of the maximum values, for fixed times the distribution in the domain of pore fluid pressures, displacements, etc…).
Author Response
Dear Sir\Madam:
Thank you very much for reviewing the manuscript “Numerical Study on the Seismic Response of Fluid-Saturated Porous Media Using the Precise Time Integration Method”, which was submitted to the Applied Sciences and giving important comments which are very beneficial to the paper improvement. All the comments are well accepted and addressed in the revised manuscript. The revisions are highlighted with red in the revised manuscript. Followed please find the detailed response to each comment.
(1) In section 4, it is used a square domain (80m, 80m) for a discretization considering a size of the finite element meshes of 8m by 8m (Loma Pietra earthquake record).
In section 5, it is used a rectangular domain (100m, 50m) for a discretization considering a size of the finite element meshes of 5m by 5m (Ninghe earthquake record).
If these choices had been made as a consequence of an optimization process, this process should be presented in the paper.
Ans: Accepted.
In the current study, the finite element mesh size is decided from the associated consideration for the computational accuracy and effort. High computational accuracy can be gained using relatively small mesh size, but this will lead to the increase of the number of finite element meshes. The computational effort and computer memory requirement will be enormous. This has been addressed in the revised manuscript (line 226-230, 284-286).
(2) In the section 4 (table 1) the material properties used in the analyses are presented. In the section 5, the material properties used in the analyses are not exhaustively presented. They should be fully presented.
Ans: Accepted.
The material properties used for the sensitivity analysis of the permeability coefficient, elastic modulus of solid skeleton and porosity have been fully presented in the revised manuscript. Please see the Table 2 to Table 4 in the section 5 of the revised manuscript.
(3) The sensitivity analysis had considered only the variability of permeability coefficient and of porosity assuming constant values of the Lame constant and of shear modulus on the basic assumptions of section 2.1 (compressibility of pore fluid, etc…). The pore fluid pressure time history can be strongly influenced by variability of other parameters. Even these choices should be presented.
Ans: Accepted.
The sensitivity of the dynamic response of fluid-saturated porous media, including the pore fluid pressure time history, to the elastic modulus of solid skeleton has also been investigated. The corresponding calculating results and conclusions have been presented in the section 5.2 of the revised manuscript.
(4) The analyses results are not presented exhaustively, only time history of two points of domain are selected. You could use a more effective representation of the results (for example: vertical and horizontal profiles of the maximum values, for fixed times the distribution in the domain of pore fluid pressures, displacements, etc…).
Ans: Accepted.
The numerical results of solid skeleton displacement distribution in different depth of the calculating domain for a fixed time have been added to the paper in the section of parametric sensitivity investigation. Please see the Figs.17, 18, 23, 24 and 29, 30 in the revised manuscript. The analysis to these numerical results have also been presented in the revised manuscript.

Reviewer 2 Report
Title: Numerical Study on the Seismic Response of Fluid Saturated Porous Media Using the Precise Time Integration Method
Important points of this paper are
a) For the liquid saturated soil, the dynamic responses are investigated considering a specific time integration scheme.
Following comments associated with the paper are presented:
· There are several time integration schemes. In this regard, the best text book is as follows
Wood, W. L. (1990). Practical time-stepping schemes, Oxford University Press, New York.
For the two phases medium (solid and liquid), the coupled hydro-mechanical response considering the poro-elasticity are well known. To the best of my knowledge best text books are
Coussy, O. (2004). Poromechanics, John Wiley and Sons, England.
Lewis, R. W., and Schrefler, B. A. (1998). The Finite Element Method in the Static and Dynamic Deformation and Consolidation of Porous Media, John Wiley & Sons, New York.
Verrujit, A. (1995). Computational Geomechanics, Kluwer Academic Publishers
· Last couple of decades the Weak formulation considering the Galerkin solution are also well established. In this regard, there are also lots of text books.
· After reading the paper, I did not find novelty/originality which can advance new knowledge to the engineering community.
· All on a sudden, authors jumped in the coupled formulation. But stepwise governing equations are essential, which will provide the insight view of the originality of the work.
· If authors want to demand, in this paper, they used known numerical solution to obtain an applied solution. Then, bunch of equations are not essential. Appropriate citations will serve the purpose. Moreover, the solid-fluid coupling constitutive relations are also absent.
· Authors separated the shape function for the solid field and fluid field. This is a good approach.
· If there is no new knowledge gap, the importance of parametric sensitivity/ uncertainty analysis/ perturbed analysis is not important. To my understanding, important part is to present the comparison of the numerical result with real experiment/ analytical solution. In this regard, for analytical solution above mentioned books are good resources. In addition, authors also may look following reference
Jeremic, B. 2018. Nonlinear Finite Elements: Modeling and Simulation of Earthquakes, Soils, Structures and their Interaction. http://sokocalo.engr.ucdavis.edu/~jeremic/LectureNotes/
· Finally, Introduction, Literature Review need to be revised to justify the novelty of the work. Stepwise governing equations and constitutive relations are also essential. Comparison of the numerical results with the analytical solution need to be presented.
Author Response
Dear Sir\Madam:
Thank you very much for reviewing the manuscript “Numerical Study on the Seismic Response of Fluid-Saturated Porous Media Using the Precise Time Integration Method”, which was submitted to the Applied Sciences and giving important comments which are very beneficial to the paper improvement. All the comments are well accepted and addressed in the revised manuscript. The revisions are highlighted with red in the revised manuscript. Followed please find the detailed response to each comment.
(1) All on a sudden, authors jumped in the coupled formulation. But stepwise governing equations are essential, which will provide the insight view of the originality of the work.
Ans: Accepted.
The stepwise governing equations for the u-p dynamic formulation have been presented in the revised manuscript (line 156-158).
(2) If authors want to demand, in this paper, they used known numerical solution to obtain an applied solution. Then, bunch of equations are not essential. Appropriate citations will serve the purpose. Moreover, the solid-fluid coupling constitutive relations are also absent.
Ans: Accepted.
As mentioned by the reviewer, the derivation of the dynamic equations of fluid-saturated porous media is not the original work of the current study. In the current study, a time-stepping explicit algorithm for the numerical solution to the dynamic equations of fluid-saturated porous media is presented. For the consideration of context integrality, the dynamic equations are presented as the basis of the algorithm presentation. The corresponding reference citation has been added to the revised manuscript (line 118). The solid-fluid coupling constitutive relations have also been presented in the revised manuscript (line 121-123).
(3) To my understanding, important part is to present the comparison of the numerical result with real experiment/ analytical solution. Comparison of the numerical results with the analytical solution need to be presented.
Ans: Accepted.
The comparison of the numerical results with the analytical solution has been presented by a numerical example. Please see the section 3.2, algorithm validation, of the revised manuscript. The comparison indicates well agreement between the numerical results and the analytical solution.

Round 2
Reviewer 1 Report
In the sections 1, 2 and 3, the presentation of background and the dynamic formulation is concise and clear.
The sections 4, and 5 have been significantly improved, even if the results in terms of pore fluid pressure and dispacements are represented only along two horizontal alignments (horizontal profiles of the maximum values).
You could use a more effective representation of the results using vertical profiles, for fixed times, of the maximum values of pore fluid pressures and displacements.
The paper can be accepted after minor revision (following the suggestion).
Author Response
Dear Sir\Madam:
Thank you very much for reviewing the revised manuscript and giving important comments which are very beneficial to the paper improvement. All the comments are well accepted and addressed in the manuscript revised for the second time. The revisions are highlighted with red in the revised manuscript. Followed please find the detailed response to each comment.
The sections 4, and 5 have been significantly improved, even if the results in terms of pore fluid pressure and displacements are represented only along two horizontal alignments (horizontal profiles of the maximum values).
You could use a more effective representation of the results using vertical profiles, for fixed times, of the maximum values of pore fluid pressures and displacements.
Ans: Accepted.
The numerical results of the vertical distribution of solid skeleton displacement for fixed times have been added to the paper in the section of parametric sensitivity investigation. Please see the Figs.19, 26 and 33 in the revised manuscript. The analysis to these numerical results have also been presented in the revised manuscript.
Reviewer 2 Report
Accepted in the present form
Author Response
Dear Sir\Madam:
Thank you very much for reviewing the revised manuscript and advising it accepted in the present form.